# The Reaction Products of the Al–Nb–B_2_O_3_–CuO System in an Al 6063 Alloy Melt and Their Influence on the Alloy’s Structure and Characteristics

**DOI:** 10.3390/ma15248898

**Published:** 2022-12-13

**Authors:** Chenggong Zhang, Min Ao, Jingyu Zhai, Zhiming Shi, Huimin Liu

**Affiliations:** School of Materials Science and Engineering, Inner Mongolia University of Technology, Huhhot 010000, China

**Keywords:** Al–Nb–B_2_O_3_–CuO reaction system, 6063 aluminum alloy, in situ NbB_2_–Al_2_O_3_ particles, aluminum matrix composites

## Abstract

To meet aero-engine aluminum skirt requirements, an experiment was carried out using Al–Nb–B_2_O_3_–CuO as the reaction system and a 6063 aluminum alloy melt as the reaction medium for a contact reaction, and 6063 aluminum matrix composites containing in situ particles were prepared with the near-liquid-phase line-casting method after the reaction was completed. The effects of the reactant molar ratio and the preheating temperature on the in situ reaction process and products were explored in order to determine the influence of in situ-reaction-product features on the organization and the qualities of the composites. Thermodynamic calculations, DSC analysis, and experiments revealed that the reaction could continue when the molar ratio of the reactants of Al–Nb–B_2_O_3_–CuO was 6:1:1:1.5. A kinetic study revealed that the Al thermal reaction in the system produced Al_2_O_3_ and [B], and the [B] atoms interacted with Nb to generate NbB_2_. With increasing temperature, the interaction between the Nb and the AlB_2_ produced hexagonal NbB_2_ particles with an average longitudinal size of 1 μm and subspherical Al_2_O_3_ particles with an average longitudinal size of 0.2 μm. The microstructure of the composites was reasonably fine, with an estimated equiaxed crystal size of around 22 μm, a tensile strength of 170 MPa, a yield strength of 135 MPa, an elongation of 13.4%, and a fracture energy of 17.05 × 10^5^ KJ/m^3^, with a content of 2.3 wt% complex-phase particles. When compared to the matrix alloy without addition, the NbB_2_ and Al_2_O_3_ particles produced by the in situ reaction had a significant refinement effect on the microstructure of the alloy, and the plasticity of the composite in the as-cast state was improved while maintaining higher strength and better overall mechanical properties, allowing for industrial mass production.

## 1. Introduction

Particle-reinforced aluminum matrix composites, widely utilized in aerospace, rail transportation, electronic packaging, and other high-tech areas, are created through addition of ceramic particles into an aluminum alloy matrix as a reinforcing phase to strengthen the matrix [1,2,3,4,5]. The in situ autogenous approach may develop reinforcing phases with small sizes, regular forms, high stiffness, and good interfacial connection with the aluminum alloy matrix [6,7]. The ceramic particles should have excellent properties, such as hardness, melting point, elastic modulus, and mismatch. TiB_2_, Al_2_O_3_, SiC, and other ceramic phases are often employed [8,9]. Ceramic particles injected into the aluminum alloy melt can operate as matrix nucleation masses, increasing the amount of non-uniform nuclei and playing an important role in grain refinement [10,11]. Mohammad, S.E. et al. [12] employed the MA approach for in situ synthesis of NbB_2_–Al_2_O_3_ nanocomposite powders; the reaction began after 10 h of grinding, with a considerable decrease in agglomerates and particle size after 30 h of grinding. Jin S.B. et al. [13] used the MA-SHS method to make NbB_2_–Al composites from Al powder, Nb powder, and B powder, and determined that the growth morphology of NbB_2x_ particles developed with the range of stoichiometric ratio 2× values (1.86–2.34) from low to high as follows: polyhedral, hexagonal, subspherical, and spherical. The mixed-salt approach for preparing particle-reinforced composites has received significant attention [14]. Li, Z. et al. [15] generated submicron TiB_2_ and NbB_2_ particles in situ by adding Nb, KBF_4_, and Ti_2_BF_6_ powders to the aluminum matrix, and the tensile strength (246 MPa) and yield strength (220 MPa) of the NbB_2_-reinforced composites were 20% and 12.8% higher, respectively, than those of the matrix alloy (205 MPa, 195 MPa); these properties were also better than those of the TiB_2_-reinforced composite (237 MPa, 223 MPa). Ding, J.H. et al. [16] employed Al, Nb, and B_4_C powders to create NbB_2_ and NbC ultra-high-temperature complex particles in situ in an Al–Cu–Mn alloy, and the composites’ tensile strength and elongation were 514 MPa and 11.5%, respectively. Jiang, J.C. et al. [17] produced NbB_2_ and Al_2_O_3_ particles in situ in a melt of A356 aluminum alloy, and the wear rate of the composite was lowered from 2.091 wt.% to 0.882 wt.% when 2.5 wt.% prefabricated blocks were added, significantly improving the composite’s wear resistance. The additive method, the intermediate alloy method, the mixed-salt method, and other methods are commonly used to prepare particle-reinforced aluminum matrix composites. The external-addition method is simple and inexpensive but prone to particle contamination. The intermediate-alloy method can control particle content in small amounts, but can also produce large-sized intermetallic compounds that cut the matrix, and intermediate alloys also have a high impurity element content. The mixed-salt approach is gentle, safe, and cost-effective; however, the resultant in situ particles are prone to agglomeration, and KBF_4_ is corrosive. In conclusion, preparation of in situ particle-reinforced aluminum matrix composites via the aluminothermic reaction between metal monomer powder and metal/non-metal oxide powder has the advantages of energy savings, uniform particle dispersion, and more, and can be used on an industrial scale.

In this research, the reaction system of Al–Nb–B_2_O_3_–CuO was employed to form NbB_2_–Al_2_O_3_ complex-phase particles in a 6063 aluminum alloy melt via an in situ contact reaction. The composite specimens were made using near-liquid-phase line casting. In order to provide a theoretical basis and data support for preparation of in situ ceramic particle-reinforced aluminum matrix composites via in situ contact reaction, the reaction mechanism of the Al–Nb–B_2_O_3_–CuO system in the alloy, the characteristics of the reaction products (morphology, size, distribution in the alloy, etc.), and the effects of those characteristics on the organization and the properties of the alloy were investigated.

## 2. Materials and Methods

The thermodynamics of the various reactions in the Al–Nb–B_2_O_3_–CuO reaction system were investigated. The differential thermal analysis method was used to examine the DSC curves of the Al–B_2_O_3_ and Al–Nb–B_2_O_3_–CuO reaction systems, and reactions that happened at the appropriate temperatures were judged by their heat absorption/exhaustion peaks. We sought to clarify whether the heat given off by the reaction-system reaction fulfilled the self-propagating reaction requirement, i.e., T_ad_ > 1800 K [18]; if the reaction was conducted under adiabatic circumstances, disregarding heat loss; and whether the reaction happened entirely according to the stoichiometric ratio.

Al powder (700 mesh, 99.9% purity), Nb powder (700 mesh, 99.9% purity), B_2_O_3_ powder (700 mesh, 99.9% purity), and CuO powder (700 mesh, 99.9% purity) were the raw materials employed in the reaction system. Figure 1 depicts the appearance morphologies of the reactants, revealing that the Al powder was almost spherical, the Nb powder was spherical, and the B_2_O_3_ powder and the CuO powder were unevenly shaped. The larger the reaction powder mesh, the smaller the powder particle size, the greater the specific surface area of the powder, the greater the surface energy, and the more violent and sufficient the reaction. Furthermore, the Al powder and CuO powder were spherical; spheres have the greatest specific surface area, resulting in particles with high surface energy. The irregularly shaped B_2_O_3_ and CuO particles had relatively large lengths and diameters; during the grinding process, irregularly shaped powder particles are more easily broken so that the size decreases and the anisotropy of the mixed powder increases, which is conducive to an adequate reaction. The influence of reactant particle size on the in situ reaction process will be investigated further in future research.

Figure 2 depicts the process roadmap for preparing composites using the in situ reaction plus near-liquid-phase line-casting approach. First, Al powder, Nb powder, B_2_O_3_ powder, and CuO powder were weighed in a grinding dish according to the molar ratio (Al:Nb:B_2_O_3_:CuO = 6:1:1:1.5) and mass ratio (Al:Nb:B_2_O_3_:CuO = 2.31:1.33:1:1.71), and the powders were well-mixed. At a pressure of 10 MPa, a desktop tablet press was used to press the combined powder into prefabricated blocks of 25 mm × 3 mm. Each prefabricated block was forced into the molten 6063 aluminum alloy with crucible tongs when the alloy was heated to 895 °C in a medium-frequency induction melting furnace. After 30 min of stirring and resting, the melt was poured into a square mold (60 mm × 3 mm × 1.5 mm) when it dropped to near the liquid-phase line of the 6063 aluminum alloy at 655 °C in order to finish the preparation procedure for the aluminum-matrix-composite specimens.

Because the content of the generated particles in the composite material was less than 5%, the specific physical phase of the block was difficult to derive with XRD analysis. A small piece of the composite-material specimen was taken and placed in a beaker with concentrated hydrochloric acid to etch the aluminum matrix (12 h), the cloudy sample obtained was rinsed with deionized water, and finally, in situ particle powder was obtained. The particle morphology, size, and distribution of the reaction products in the 6063 aluminum alloy matrix were observed with a field-emission scanning electron microscope (FEI QUANTA 650 FEG), the grain size of each composite was tested with Nano measure software 1.2, and the tensile strength of each material was tested with a universal material testing machine.

## 3. Results and Discussion

### 3.1. Thermodynamic Analysis

Table 1 shows the potential reactions, (1)–(5), in the Al–Nb–B_2_O_3_–CuO system, as well as each reaction’s Gibbs free energy as a function of temperature. The fundamental thermodynamic characteristics of the reaction system under normal circumstances were questioned using the handbook of inorganic thermodynamics, and the enthalpy and entropy changes of the reactants and products at ambient temperature, ambient pressure, and T temperature were calculated using Equations (6)–(9). The Gibbs free energy values of reactions (1)–(5) were calculated using the Gibbs free energy equation, Equation (10), and the relationship between the Gibbs free energy criteria and 0 was utilized to assess whether or not the reaction would continue spontaneously.
materials-15-08898-t001_Table 1Table 1The relationship between Gibbs free energy and temperature [17,18,19].Chemical ReactionGibbs Free Energy (J/mol)Number2Al + 3CuO → Al_2_O_3_ + 3CuΔGTθ = −1,207,712 + 34.16 T(1)2Al + B_2_O_3_ → Al_2_O_3_ + 2[B]ΔGTθ = −404,844 + 12.34 T(2)Nb + 2[B] → NbB_2_ΔGTθ = −251,040 + 22.35 T(3)AlB_2_ + Nb → NbB_2_ + AlΔGTθ = −184,096 + 5.15 T(4)Al + 2[B] → AlB_2_ΔGTθ = −66,944 + 40.809 T(5)
(6)$$\Delta H_{T}^{\theta }=\Delta H_{298}^{\theta }+\int_{298}^{T}\Delta C_{P}dT$$
(7)$$\Delta S_{T}^{\theta }=\Delta S_{298}^{\theta }+\int_{298}^{T}\frac{\Delta C_{P}}{T}dT$$
(8)$$\Delta H_{298}^{\theta }={\sum \left(\Delta H_{298}^{\theta }\right)}_{products}-{\sum \left(\Delta H_{298}^{\theta }\right)}_{reac\tan ts}$$
(9)$$\Delta S_{298}^{\theta }={\sum \left(\Delta S_{298}^{\theta }\right)}_{products}-{\sum \left(\Delta S_{298}^{\theta }\right)}_{reac\tan ts}$$
(10)$$\Delta G_{T}^{\theta }=\Delta H_{T}^{\theta }-T\Delta S_{T}^{\theta }$$

Here, $\Delta H_{298}^{\theta }$ is the system enthalpy change at the standard temperature; $\Delta S_{298}^{\theta }$ is the system entropy change at the standard temperature; $\Delta C_{P}$ is thermal fusion; $\Delta H_{T}^{\theta }$ is the system enthalpy change at T temperature; $\Delta S_{T}^{\theta }$ is the system entropy change at T temperature; and $\Delta G_{T}^{\theta }$ is reaction-system Gibbs free energy.

Figure 3 depicts the Gibbs free energy values of Table 1 obtained from the query, plotted versus temperature. As shown in Figure 3, the Gibbs free energy of reactions (1)–(5) was ΔG < 0. The free energy of reactions (1), (3), and (5) increased little as the temperature increased, and thermodynamic analysis suggested that reactions (1)–(5) were achievable under these test circumstances at temperatures below 1200 °C. These reactions occurred more easily when the Gibbs free energy of the reaction was smaller. The system reaction 2Al + 3CuO → Al_2_O_3_ + 3Cu was the easiest-occurring, followed by the reaction 2Al + B_2_O_3_ → Al_2_O_3_ + 2[B], following which the reactions progressed according to degree of difficulty: (3)–(5).

As indicated in Figure 4, DSC analysis was undertaken on the two reaction systems Al–B_2_O_3_ and Al–Nb–B_2_O_3_–CuO. The temperature at which the reactions took place was governed by the heat absorption/exhaustion peak. The heat absorption peak at 155 °C in the Figure 4a DSC curve represents the melting process of B_2_O_3_ powder, the strong heat absorption peak at 650 °C represents the melting of Al powder, the exothermic peak at 814 °C represents the heat released by the reaction 2Al + B_2_O_3_ → Al_2_O_3_ + 2[B], and the exothermic peak at 930 °C represents the heat released by the reaction Al + 2[B] → AlB_2_. The reactions at 155 °C and 650 °C in the DSC curve of Figure 4b were the same as those in Figure 4a due to the addition of Nb powder and CuO powder in the system, which increased the chance of particle collision during the grinding process. The irregularly shaped B_2_O_3_ and CuO were broken, which increased the specific surface area between the powders, increased the surface activity of the powders, and decreased the starting temperature of the reaction. As a result, Al and B_2_O_3_ interacted at 677 °C [20]. The heat generated by the reaction Al + 2[B] →AlB_2_ was 805 °C, and the strong exothermic peak at 895 °C is the heat released by the reaction 2Al + 3CuO → Al_2_O_3_ + 3Cu, which gave out a huge quantity of heat and had a stronger peak, resulting in the succeeding reactions. Nb + 2[B] → NbB_2_ and Nb + AlB_2_ → NbB_2_ + Al occurred at 1007 °C, when the distinct exothermic peaks were muted and a single peak appeared. This is consistent with the findings of prior research [17,20,21].

The DSC test results confirmed that the criterion of high-temperature self-propagation was met at the preheating temperature, and that the adiabatic temperature, T_ad_, of the reaction was estimated using the heat balance equation, as indicated in Equation (11), given a system preheating temperature of T_0_ [18]:(11)$$\sum n_{i}{\left(H_{T_{0}}^{\theta }-H_{298}^{\theta }\right)}_{i,reac\tan ts}-\Delta H_{298}^{\theta }=\sum n_{i}{\left(H_{T}^{\theta }-H_{298}^{\theta }\right)}_{i,products}\;,$$
where n is the coefficient of the equation, $H_{T_{0}}^{\theta }$ is the standard molar enthalpy of the reaction, and $H_{T}^{\theta }$ is the standard molar enthalpy of production.

According to thermodynamic calculations and DSC analysis, given a system with a preheating temperature of T_0_ = 895 °C, an adiabatic temperature for reaction (2) of T_ad_ ≈ 3400 K > 1800 K, and a reaction (1) adiabatic temperature of T_ad_ ≈ 3200 K > 1800 K, the reaction was a thermal explosion reaction, releasing a large amount of heat that was sufficient to sustain the reaction and meet the high-temperature self-propagation conditions. The heat generated by aluminothermic processes (1) and (2) raised the temperature of the Al–Nb–B_2_O_3_–CuO system to more than 3000 K in an instant. Reactions (3)–(5) were intermediate reactions, and the plot of Gibbs free energy against temperature showed that ΔG < 0 processes could have occurred. As a result, it was demonstrated thermodynamically that NbB_2_ and Al_2_O_3_ ceramic particles may be produced utilizing the Al–Nb–B_2_O_3_–CuO system as the reactant.

### 3.2. Mechanistic Analysis of the In Situ Reaction Process

The goal of this kinetic study was to evaluate the functions of Al, Nb, B_2_O_3_, and CuO, as well as the reaction process of the Al–Nb–B_2_O_3_–CuO system and the effect of each element on the in situ reaction during high-temperature self-propagation. According to the previous thermodynamic calculations and DSC analysis of the reaction system, when the molar ratio of the components in the reaction system was Al–Nb–B_2_O_3_–CuO = 6:1:1:1.5, the exotherm of the Al–CuO reaction in the early stage of SHS greatly promoted complete progress of the self-propagation reaction and decomposition of AlB_2_ at high temperatures. The inclusion of B_2_O_3_ increased the wettability of the B and aluminum alloy melts, allowing for more uniform particle dispersion.

The reaction mechanism of the Al–Nb–B_2_O_3_–CuO system in the self-propagating high-temperature synthesis process was simulated based on the foregoing study, as shown in Figure 5.

Stage 1: When the prefabricated block is immersed in the aluminum melt, the Al powder melts and is disseminated among the other reactants, as illustrated in Figure 5a. The DSC analysis in Figure 4b shows that the interaction of Al with B_2_O_3_ initially produces Al_2_O_3_ particles, and that the heat generated from that reaction drives the reaction of Al with CuO. In CuO and B_2_O_3_, Al atoms replace Cu and B atoms, and the replacement B atoms are substantially more reactive than was the single B, resulting in reactive [B] atoms. The temperature within the system rises instantly after the Al–heat reaction occurs, at which time the Nb has a high solubility in the Al, forming an Nb-rich region, and the highly wettable [B] moves toward the Nb-rich region. Once the critical [B] concentration is reached, the NbB_2_ nucleates, as shown in the transition from (b) to (c) in Figure 5.

Stage 2: Because of [B]’s high activity and strong wettability with Al, the interaction of Al atoms with [B] produces the AlB_2_ intermediate product, as illustrated in Figure 5d. As the melt temperature increases, the AlB_2_ intermediate product becomes unstable, interacting with Nb to generate NbB_2_ with a stable structure until AlB_2_ is exhausted, as seen in Figure 5e.

Stage 3: Following the completion of the self-propagation reaction, the melt temperature is decreased to the preheating temperature, and the NbB_2_ and Al_2_O_3_ particles produced in situ are uniformly distributed in each corner of the melt, as illustrated in Figure 5f. It was possible to create in situ NbB_2_- and Al_2_O_3_-particle-reinforced aluminum matrix composite melts.

In conclusion, the thermodynamic calculations and the kinetic study of the Al–Nb–B_2_O_3_–CuO system showed that this reaction could take place and that NbB_2_ and Al_2_O_3_ particles were formed and dispersed in the 6063 aluminum alloy matrix.

Figure 6 depicts the XRD pattern of the sample after acid etching. After acid etching, the predominant phase compositions in the powder were NbB_2_ particles produced in situ (lattice parameter a (c) = 0.31 (0.33) nm) and α-Al_2_O_3_ particles (lattice parameter a (c) = 0.47 (1.29) nm), both of which corresponded to the hexagonal crystal structure [22]. NbB_2_ particles were formed from the reactions Nb + 2[B] → NbB_2_ and AlB_2_ + Nb → NbB_2_ + Al, whereas Al_2_O_3_ particles were formed from the reactions 2Al + B_2_O_3_ → Al_2_O_3_ + 2[B] and 2Al + 3CuO → Al_2_O_3_ + 3Cu. As the interaction of Al and B_2_O_3_ took place preferentially in the system, the heat emitted stimulated the reaction of Al with CuO, raising the preheating temperature, T_0_, of reaction (2), as well as the system temperature [23]. As the temperature rose, the amorphous alumina underwent a continuous phase transition, i.e., finally converting into stable α-Al_2_O_3_ [20]. Because of the large amount of heat released by the in situ reaction, the excess phase Al_2_O_3_ generated by the in situ reaction 2Al + B_2_O_3_ → Al_2_O_3_ + 2[B] transformed into the α-phase at high temperatures, and the Al_2_O_3_ generated by the reaction 2Al + 3CuO → Al_2_O_3_ + 3Cu was α-type [22,24], so the crystalline forms of Al_2_O_3_ in the material phases tested by XRD were all α-type.

### 3.3. Distribution of In Situ Particles in the Alloy

Figure 7 depicts the scanning examination of the composite with a 2.3 wt.% in situ particle content (1 wt.% NbB_2_, 1.3 wt.% Al_2_O_3_). Figure 7a depicts the particle distribution in the alloy, revealing that the great majority of the particles were scattered within the α-Al grains, which suggests intracrystalline dispersion. Intergranular distribution occurs when a few particles are scattered along the grain borders of α-Al grains and trigonal grain boundaries. Because of van der Waals attraction, submicron Al_2_O_3_ particles always tend to aggregate into clusters in the metal matrix, and the strengthening effect is dramatically diminished when the size of the clusters produced by the Al_2_O_3_ particles reaches microns or more [25]. Dual-scale and dual-particle-reinforced aluminum matrix composites have considerably enhanced plasticity while keeping greater strength and have better overall mechanical characteristics when compared to single-scale and single-phase particle-reinforced aluminum matrix composites. To overcome the aggregation problem of Al_2_O_3_ particles or to reduce the degree of aggregation of the Al_2_O_3_ particles, a more effective method, used in this experiment, was to use multi-scale reinforcing phase particles, i.e., dual-scale submicron (Al_2_O_3_) and micron (NbB_2_) particles, to co-reinforce the aluminum matrix composites. The Al_2_O_3_ particles were better-diffused and -distributed throughout the crystal under the action of NbB_2_ particles, and only a tiny number of NbB_2_ particles were scattered along the grain borders. In fact, α-Al tended to stick to the surfaces of the second-phase particles for non-uniform nucleation in order to lower the free energy of the system during the crystallization process [26]. The particles inside the grains encouraged nucleation, and the resulting microstructure was more homogeneous and finer. The smaller the grain size, the greater the strength and the hardness, as well as the plasticity and the toughness. This is because the finer the grain is, the more uniformly plastic deformation can be dispersed in more grains, resulting in a smaller concentration of internal stress, and finer grain means more crystal interface, more curved grain boundaries, more opportunities for interlocking grains and grains, more unfavorable crack propagation and development, tighter grain proximity with one another, and better strength and toughness.

Figure 7b depicts an expanded view of Figure 7’s region A. When the two pictures (a,b) are combined, it is clear that the majority of the particles were still diffusely dispersed throughout the matrix, with just a few seeming to be agglomerated. The face-scan analysis of Figure 7b, along with the distribution of Nb elements in (c) and the distribution of B elements in (d), led us to the conclusion that the reaction created NbB_2_ particles with a hexagonal shape and an average longitudinal size of 1 μm. The reaction generated approximately spherical Al_2_O_3_ particles with an average size of 0.2 μm, based on the distribution of Al elements in Figure 7e and of O elements in (f) coupled with XRD analysis. SEM inspection of the composites revealed NbB_2_ and Al_2_O_3_ particles, indicating that the reaction could form hard ceramic particles in the alloy. SEM revealed that the in situ NbB_2_ and Al_2_O_3_ particles were randomly aggregated together and that these aggregated particles were not all strongly connected to one another. These nearby particles subsequently built channels between themselves, and the linked particles obstructed solute atom transport in the liquid phase. The channels between the particles limited the transport route of solute atoms and hindered solute-atom diffusion. The advantages of in situ composite preparation are as follows: (1) Through rational selection of chemical-reaction components and circumstances, the type, size, amount, and distribution of in situ-produced enhancers may be regulated; (2) In situ-created ceramic reinforcement is thermodynamically stable and eliminates the issue of poor compatibility between applied ceramic particles and substrates, as well as having strong bonding with substrates; (3) Reinforcement is created with an in situ chemical reaction, which eliminates the phases of separate synthesis, processing, and inclusion of the reinforcement, resulting in a simpler process and lower production costs; (4) In situ-created reinforcements in liquid metals can be employed for casting production and creation of net proximal components with complicated forms and huge dimensions. The in situ particles were incorporated in the alloy matrix and had strong interfacial bonding. They spontaneously assembled at the growth interface of the α-Al, preventing solute atom migration to the growth interface and blocking grain development, which resulted in grain refinement.

### 3.4. Effect of In Situ Particles on the Grain Structure and Properties of Alloys

Figure 8 depicts the microstructures and the average grain sizes of in situ-reaction near-liquid-phase line-casting aluminum matrix composites under various particle-content circumstances. Figure 8a depicts the microstructure of the basic 6063 aluminum alloy cast state; at this point, the microstructure was coarser, and the grains had a rose-like appearance [27]. The microstructure of the composites evolved from relatively coarse dendrites to fine near-equiaxial crystals as the concentration of complex-phase particles increased, as illustrated in Figure 8a,c. When the particle content was 1.2 wt.% (0.5 wt.% NbB_2_, 0.7 wt.% Al_2_O_3_), the microstructures of the composites were significantly refined. When the particle content was increased further to 2.3 wt.% (1 wt.% NbB_2_, 1.3 wt.% Al_2_O_3_), the microstructures of the composites were even finer (Figure 8c), with uniformly fine sub-equiaxial crystals, and the average grain size reached 22 μm, as shown in Figure 8g. When the particle content increased to 3.5 wt.% (1.5 wt.% NbB_2_, 2 wt.% Al_2_O_3_), more heat was released from the reaction, and the reaction process became more violent, resulting in melt sputtering and a decrease in the actual particle content in the melt; therefore, the grain size coarsened [17] and the average grain size increased to 30 μm, as shown in Figure 8h. The in situ-created NbB_2_ and Al_2_O_3_ particles operated as nucleation masses during solidification, increasing the number of non-uniform nuclei and decreasing the amount of subcooling required for solidification, which helped to refine the grains [28,29,30,31,32,33]. The in situ NbB_2_ and Al_2_O_3_ particles were randomly dispersed along the grain borders, thereby impeding grain boundary movement and preventing grain expansion. Simultaneously, when the alloy was prepared using the near-liquid-phase line-casting method, the proximally ordered quasi-solid-phase atomic group acted as a heterogeneous nucleation substrate due to the lower casting temperature, increasing the number of nuclei and further refining and homogenizing the grains [34]. As a result, when the compound phase particles were introduced at 2.32 wt.%, the alloy’s microstructure was successfully refined, and the composite’s solidification organization was finer and more uniform.

Figure 9 depicts the composite characteristics at various NbB_2_–Al_2_O_3_ particle concentrations. In the cast condition, the tensile strength of the 6063 aluminum alloy was 65 MPa, the yield strength was 40 MPa, the elongation was 4.8%, and the fracture energy was 2.21 × 10^5^ KJ/m^3^. The tensile strength, yield strength, elongation, and fracture toughness of the composites rose as the complex-phase particles’ NbB_2_–Al_2_O_3_ concentration increased. When the complex-phase particles’ NbB_2_–Al_2_O_3_ content was 2.3 wt.%, the composites’ tensile strength reached 170 MPa, the yield strength reached 135 MPa, the elongation reached 13.4%, and the fracture energy reached 17.05 × 10^5^ KJ/m^3^, respectively: increases of 161.5%, 237.5%, 179%, and 671%, respectively, when compared to those qualities in the 6063 aluminum alloy. The Hall–Petch relationship [35] governs the link between grain size and metallic material strength, since a specific mass proportion of complex particles has a grain-refining impact on a given matrix.

The smaller the grain size of the metal material was, the more grain boundaries there were, and the more grains there were in various dislocations, the better the plastic deformation resistance was; therefore, the strength was higher when coupled with the microstructure of the composite material in Figure 8. At the same time, grain refinement occurred, reducing the density of the dislocations dispersed in each grain and allowing the material to undergo large plastic deformation without causing a large concentration of stress, which would have caused the material to crack. Dislocations were prevented from moving so as to be able to absorb a large impact; this is expressed as high toughness. According to fine-grain-strengthening theory, the pair of contradictory relationships between the strength and the toughness of the material can be effectively resolved only through the process of continuous grain refinement of metal materials, and strength and toughness grow simultaneously in the design concept of high-strength and high-toughness materials.

The dislocations were hindered when passing through the second phase under the applied load from the Orowan strengthening mechanism [35], and when the second phase was a hard particle that was not easily deformed, the dislocations continued to move after bypassing the second phase, forming a dislocation ring around the hard particle. The greater the number of dislocation lines that interacted with the second phase, the more dislocation loops left behind, giving reinforcement.

According to load-transfer theory [30], load transfer happens between the matrix and the reinforcement when the load is subjected to external forces and transferred from the soft phase to the hard phase, improving the material’s strength. Furthermore, throughout the deformation process, dislocations multiply and entangle with one another, resulting in process hardening, which raises the material’s strength even further.

When the theoretical value of the complex-phase particle NbB_2_–Al_2_O_3_ content was 3.5 wt.%, the actual particle content in the composite was less than the theoretical value due to melt sputtering, but the number of particles in the composite was still greater than that at a higher content of 2.3 wt.%, at which point the composite’s tensile strength and hardness tended to decrease. This is because the possibility of particle agglomeration grows as the content of the complex-phase particles increases and the agglomeration phenomena between the particles becomes more severe [36]. When a composite material is subjected to external pressures, the stress concentration increases, the interfacial bonding capacity between the two particle clusters and the matrix weakens, and the microcracks continue to expand, finally leading to fractures [33,36]. As a result, the more-appropriate inclusion of the complex-phase particles was 2.3 wt.%, and the strength and toughness of the as-cast composites were enhanced.

## 4. Conclusions

The in situ-reaction near-liquid-phase wire-casting technique was used to manufacture a (NbB_2_–Al_2_O_3_)p/6063 composite. We examined, tested, and compared its standard organization, its characteristics, and its organization with a basic 6063 aluminum alloy cast near the liquid-phase line. The following findings were obtained:The aluminothermic interaction of Al–B_2_O_3_ and Al–CuO during SHS created Al_2_O_3_ particles and displaced [B] atoms, increasing the wettability of the B in the Al, which nucleated to form NbB_2_ and AlB_2_ once a threshold [B] concentration surrounding the Nb and the Al was achieved. Furthermore, at high temperatures, the exothermic reaction between the Al and the CuO promoted the interaction of the AlB_2_ with the Nb to generate NbB_2_. At the same time, due to the high solubility of [B] in this melt, the high [B] concentration promoted stoichiometric NbB_2_ production.The Al_2_O_3_ particles were dispersed throughout the crystal, while the NbB_2_ particles were primarily dispersed throughout the crystal with a tiny number near the grain boundaries. The intracrystalline particles stimulated nucleation, whereas the particles near the grain borders inhibited grain development, resulting in a very tiny microstructure.The grain size of each composite reduced and subsequently grew when the NbB_2_ and Al_2_O_3_ content of each composite increased, and when the complex-phase particle content was 2.3 wt.%, the grain size of each composite was the minimal value of 22 μm.The ultimate tensile strength, the yield strength, the elongation, and the fracture energy of the composites increased and then declined as the concentration of NbB_2_ and Al_2_O_3_ in each composite increased. When the complex-phase particle concentration was 2.3 wt.%, the ultimate tensile strength, yield strength, elongation, and fracture energy were 170 MPa, 135 MPa, 13.4%, and 17.05 × 10^5^ KJ/m^3^, respectively. These values were 161.5%, 237.5%, 179%, and 671% more, respectively, than the values of those qualities in the 6063 aluminum alloy.Under the action of double particles, the composite developed more comprehensive characteristics in the as-cast condition as the NbB_2_ and Al_2_O_3_ concentrations in the 6063 aluminum alloy increased.

## Figures and Tables

**Figure 1 materials-15-08898-f001:**
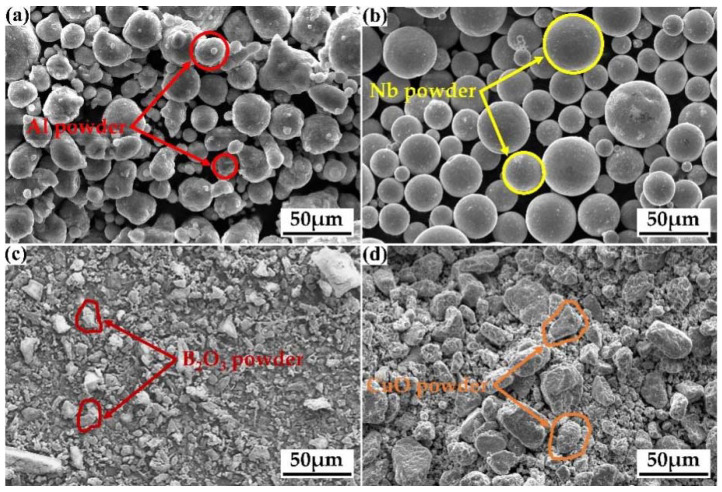
The appearances of the reactants: (**a**) Al powder, (**b**) Nb powder, (**c**) B_2_O_3_ powder, and (**d**) CuO powder.

**Figure 2 materials-15-08898-f002:**
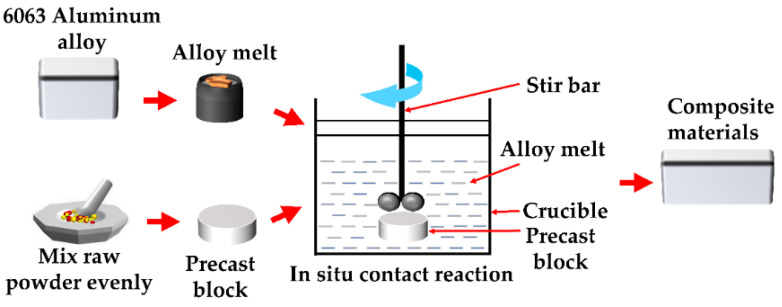
The in situ reaction and near-liquid-phase line-casting method for the composite-material preparation process.

**Figure 3 materials-15-08898-f003:**
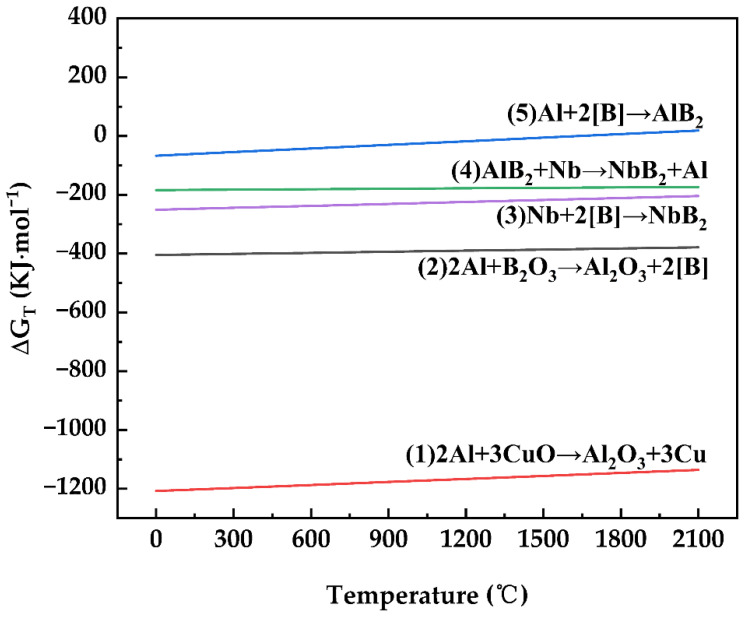
Gibbs free energy versus temperature for reactions (1)–(5).

**Figure 4 materials-15-08898-f004:**
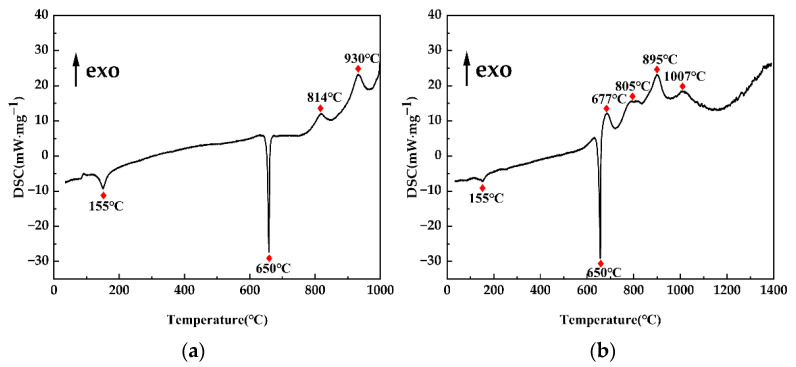
The DSC studies of the reaction systems: (**a**) Al–B_2_O_3_ and (**b**) Al–Nb–B_2_O_3_–CuO.

**Figure 5 materials-15-08898-f005:**
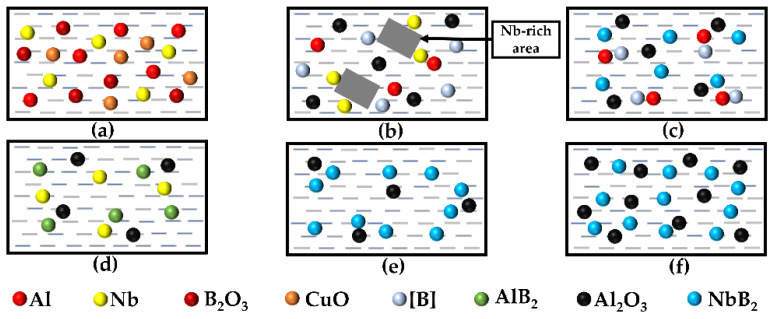
The Al–Nb–B_2_O_3_–CuO system in situ reaction mechanism: (**a**) aluminum melting and dispersion in the reactants, (**b**) the dissolution of Nb in Al to form the Al–Nb melt and the aluminothermic reaction, (**c**) the reaction of Nb with [B] to form NbB_2_ particles, (**d**) the generation of intermediate product AlB_2_, (**e**) the reaction of AlB_2_ with Nb to form NbB_2_ particles, and (**f**) in situ NbB_2_ and Al_2_O_3_ particles to strengthen the melt of aluminum matrix composites.

**Figure 6 materials-15-08898-f006:**
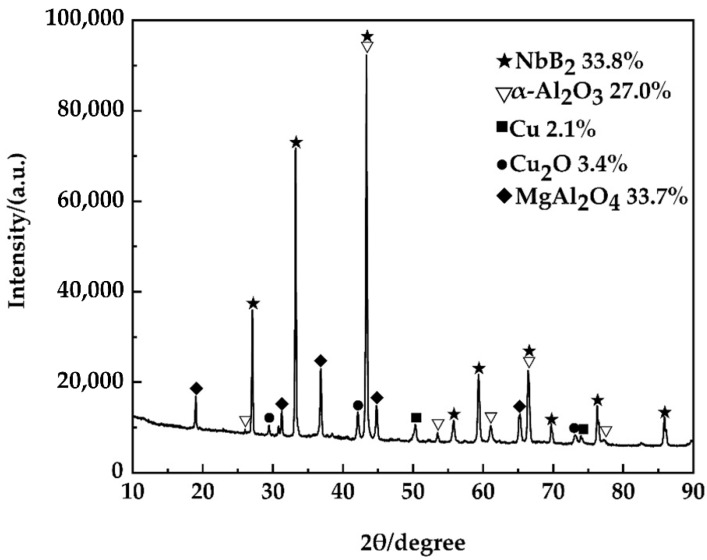
The XRD pattern of an acid-etched powder sample.

**Figure 7 materials-15-08898-f007:**
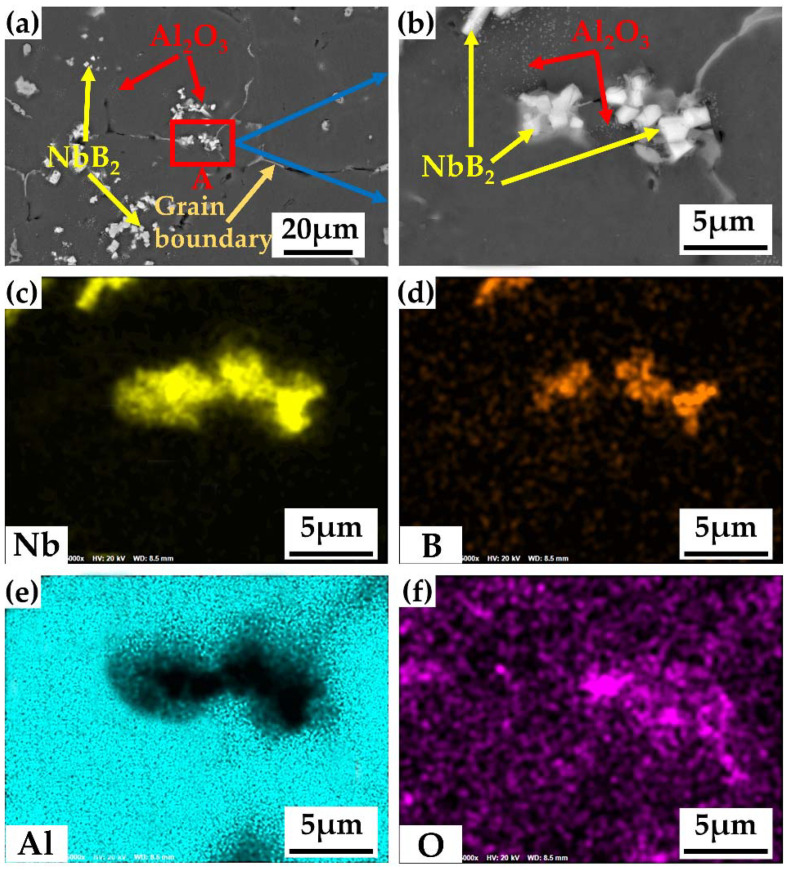
The particle morphology and a face-scan study of a composite with 2.3 wt.% particle content: (**a**) the particle morphology and distribution under a microscope, (**b**) a close-up of region A in Figure 7a, (**c**) the Nb element distribution, (**d**) the B element distribution, (**e**) the Al element distribution, and (**f**) the O element distribution.

**Figure 8 materials-15-08898-f008:**
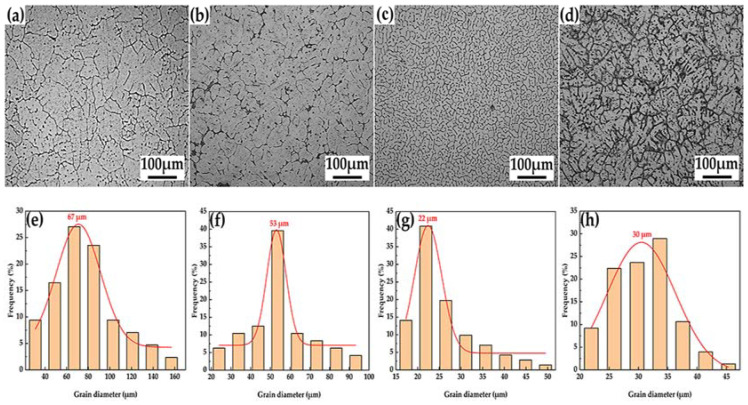
Microstructures of the composites with different particle contents: (**a**) matrix 6063 aluminum alloy, (**b**) 1.2 wt.%, (**c**) 2.3 wt.%, and (**d**) 3.5 wt.%. (**e**–**h**) Grain size distribution.

**Figure 9 materials-15-08898-f009:**
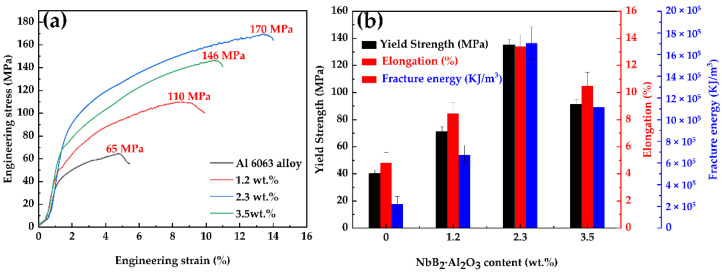
Composite properties with varying particle contents: (**a**) the stress–strain curve and (**b**) the yield strength, elongation, and toughness.

## Data Availability

The data presented in this study are available on request from the corresponding authors.

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
