# Peer review of "The Reaction Products of the Al–Nb–B2O3–CuO System in an Al 6063 Alloy Melt and Their Influence on the Alloy’s Structure and Characteristics"

_materials, 2022, doi:10.3390/ma15248898_

Round 1
Reviewer 1 Report
The article is devoted to the study of composites based on Al-Nb-B2O3-CuO, as well as an aluminum matrix, as well as to the study of the processes of structure formation in the in situ mode and the influence of the molar ratio on the properties of the resulting composites. To characterize the samples, the authors use a fairly large number of different research methods, including scanning electron microscopy, thermogravimetric analysis, thermodynamic calculations, etc. In general, this direction is quite promising and interesting not only for a narrow circle of researchers involved in such research, but also for a wide range of readers. However, before accepting this article for publication, the authors should make a number of corrections and answer the reviewer's questions.
1. In general, the presented data are quite interesting, however, in the abstract, the authors should give more details about the prospects of this study and the relevance of the practical application of these composites and their scope.
2. When analyzing the morphological features of the powders under study, the authors should pay attention to the isotropy of particle sizes, as well as their shape, which can have a rather serious effect on the processes of composite formation.
3. When analyzing X-ray diffraction patterns, the authors should pay attention to low-intensity diffraction reflections that have not been identified in any way, while reflecting the reasons for their appearance, and also indicate the ratios of the observed phases.
4. As can be seen from these morphological features of the obtained composites, the formation of inclusions in the form of NbB2 and Al2O3 is observed on the surface, while the authors do not indicate to what extent they are embedded in the composite structure. At the same time, most of these inclusions are located near cracks and grain boundaries; this should also be indicated in more detail in the text of the article.
5. The results of hardening should be compared not only between the compositions obtained, but also with other literature data, in order to determine the effectiveness of the proposed method for obtaining composites.
6. The conclusion partially duplicates the abstract, this should be eliminated.
Author Response
Dear Reviewer,
Thank you for your diligence!
In response to your valuable remarks, the following changes and clarifications have been made:
- In general, the presented data are quite interesting, however, in the abstract, the authors should give more details about the prospects of this study and the relevance of the practical application of these composites and their scope.
Thank you for your insightful comments! The abstract section of the publication clarifies the study's background, prospects, and significance of its practical applications, which are highlighted in red in the original manuscript. Please review.
- When analyzing the morphological features of the powders under study, the authors should pay attention to the isotropy of particle sizes, as well as their shape, which can have a rather serious effect on the processes of composite formation.
Thank you for your insightful comments! Each anisotropy of the shape and size of the reacting powder material is described in detail in the Materials and Methods section of the publication and highlighted in red in the original manuscript. Please review.
- When analyzing X-ray diffraction patterns, the authors should pay attention to low-intensity diffraction reflections that have not been identified in any way, while reflecting the reasons for their appearance, and also indicate the ratios of the observed phases.
Thank you for your insightful comments! As shown in Figure 6 of the original manuscript, the low-intensity diffraction peaks in the paper XRD diffractogram have been enhanced, and the physical phases have been quantified. Please review.
- As can be seen from these morphological features of the obtained composites, the formation of inclusions in the form of NbB2 and Al2O3 is observed on the surface, while the authors do not indicate to what extent they are embedded in the composite structure. At the same time, most of these inclusions are located near cracks and grain boundaries; this should also be indicated in more detail in the text of the article.
Thank you for your insightful comments! The extent of particle embedding in the composite and the location of particle presence are specified in the original publication's findings and analysis section 3.3, which is highlighted in red. Please review.
- The results of hardening should be compared not only between the compositions obtained, but also with other literature data, in order to determine the effectiveness of the proposed method for obtaining composites.
Thank you for your insightful comments! In this experiment, the reaction system was Al-Nb-B2O3-CuO and the reaction medium was 6063 aluminum alloy in order to produce in situ particulate NbB2-Al2O3 reinforced aluminum matrix composites. The method is innovative and practical, and it can be utilized in industrial mass production. Due to the poor casting qualities of the matrix 6063 aluminum alloy, the mechanical properties of the as-cast composites are not extremely high and are lower than the mechanical properties of composites manufactured with Al-Si alloy as matrix, as outlined in the manuscript [15], [20]. Nonetheless, a comparison is provided using the Al-Mg-Si alloy as the matrix, as shown in the manuscript [33]. The ultimate tensile strength of the composite in the as-cast state is 73 MPa when the complex phase particle content is 15% wt., using 6061 aluminum alloy as the matrix and adding Al2O3-SiC particles by the additive method. Based on the paper [36], ZrB2-Al2O3 particle-reinforced composites were produced in situ using AA6016 aluminum alloy as the matrix, and the composites' as-cast ultimate tensile strength was 130 MPa when the complex phase particle concentration was 3 wt.%. In the composites produced in this experiment using Al-Mg-Si alloy as the matrix and under similar circumstances, the ultimate tensile strength was 170 MPa at an in situ NbB2-Al2O3 particle concentration of 2.3 wt.%, and the mechanical parameters were superior to those described in the literature. Please review.
- The conclusion partially duplicates the abstract, this should be eliminated.
Thank you for your insightful comments! The conclusion section of the paper has been revised to reduce redundancy with the abstract, which was highlighted in red in the original text. Please review.
Have a nice day and thanks again!
Kind regards.
Chenggong Zhang

Reviewer 2 Report
Journal: Materials (ISSN 1996-1944)
Manuscript ID: Materials- 2061938
The authors presented an article on “The reaction products of the Al-Nb-B2O3-CuO system in Al 6063 alloy melt and their influence on the alloy's structure and characteristics”. I think the article is well organized and suitable for the "Materials" magazine. Studies in the literature were compared, detailed analyzes were made and satisfactory results were obtained. However, the article will be ready for publication after a minor revision. Comments are listed below.
Note: Turnitin similarity rate is 16%.
1. In the introduction, the properties of the particles added to the aluminum matrix should be mentioned.
2. The last paragraph of the introduction should clearly state the strengths of the study and how it differs from other studies.
3. The dimensions of the powder particles are given in the material method section. B2O3 powder particles appear smaller in the SEM photograph. Is there an error?
4. The powder particles used in the material method section are given proportionally. Percentages by weight can be given.
5. The authors mentioned the nucleation of particles within grain boundaries on page 8, line 252. What is the effect of nucleation on mechanical properties?
6. The article contains numerous typographic and language errors. It should be corrected.
7. The article should be rearranged by taking into account the journal writing rules and citation rules.
8. The article is well-organized, yet there is a reference problem. First, your reference list contains no article from the “Materials” journal. If your work is convenient for this journal's context, then there are many references from this journal. Secondly, cited sources should be primary ones. Namely, the indexed area shows the power of a paper and directly your paper's reliability. Please make regulations in this direction.
*** Authors must consider them properly before submitting the revised manuscript. A point-by-point reply is required when the revised files are submitted.

Author Response
Dear Reviewer,
Thank you for your diligence!
In response to your valuable remarks, the following changes and clarifications have been made:
- In the introduction, the properties of the particles added to the aluminum matrix should be mentioned.
Thank you for your insightful comments! The selection of ceramic particles to be added to the aluminum matrix is described in the document's introduction, and the attributes of the particles should be highlighted in red in the original manuscript. Please review.
- The last paragraph of the introduction should clearly state the strengths of the study and how it differs from other studies.
Thank you for your insightful comments! In the final paragraph of the paper's introduction, the advantages and disadvantages of various methodologies for studying the production of composite materials are discussed, and the differences between this research and other studies are highlighted in red in the original text. Please review.
- The dimensions of the powder particles are given in the material method section. B2O3 powder particles appear smaller in the SEM photograph. Is there an error?
Thank you for your insightful comments! The microscopic particles of B2O3 powder revealed in SEM images were error-free. Due to the fact that B2O3 powder is a non-metallic oxide and lighter in weight, the size of B2O3 particles and other particles in the reaction powder appear to be slightly smaller at the same mesh. Prior to SEM, the powder was bonded to a conductive adhesive and purged with high-purity argon gas to prevent pollution. Larger particles exhibited weaker adhesion and were blown away, resulting in the smaller B2O3 particles visible in SEM images. Due to the average size of the reaction powder, the B2O3 used in the actual reaction process has a mesh size of 700. Please review.
- The powder particles used in the material method section are given proportionally. Percentages by weight can be given.
Thank you for your insightful comments! The reaction powders are supplied in weight percentages under their ratios, which are marked in red in the original publication's Materials and Methods section. Please review.
- The authors mentioned the nucleation of particles within grain boundaries on page 8, line 252. What is the effect of nucleation on mechanical properties?
Thank you for your insightful comments! The effect on the mechanical properties of the material when particles are nucleated within the grain boundaries has been described in the manuscript in relation to the nucleation of particles within the grain boundaries, which is mentioned on page 8, line 252, and highlighted in red in the original manuscript. Please review.
- The article contains numerous typographic and language errors. It should be corrected.
Thank you for your insightful comments! The text has been corrected for typographical and grammatical errors at "https://www.mdpi.com/authors/english," as shown in the original manuscript. Additionally, please attach the embellishment certificate below. Please review.
- The article should be rearranged by taking into account the journal writing rules and citation rules.
Thank you for your insightful comments! The text has been updated to adhere to journal writing and citation guidelines. Please review.
- The article is well-organized, yet there is a reference problem. First, your reference list contains no article from the “Materials” journal. If your work is convenient for this journal's context, then there are many references from this journal. Secondly, cited sources should be primary ones. Namely, the indexed area shows the power of a paper and directly your paper's reliability. Please make regulations in this direction.
Thank you for your insightful comments! The paper has been updated with references to the study from the journal "Materials," as shown in [1,2], [4,5], and the additional references are highlighted in the original manuscript. Please review.
Have a nice day and thanks again!
Kind regards.
Chenggong Zhang

Round 2
Reviewer 1 Report
The authors answered all the questions, the article can be accepted for publication.